# The Effects of a Professor's Professionalism and Diversity on the Perception and Satisfaction of Education in the Liberal Arts Curriculum

**Mi-Young An [1], Susie Yoon [2,*] and Sang-Ho Han [3,*]**

[1] Department of Earlychildhood Education, Cheju Halla University, 38 Halladaehak-ro, Jeju-si, Jeju Special Self-Governing Province 63092, Korea; myahn@chu.ac.kr

[2] Department of Nursing, Cheju Halla University, 38 Halladaehak-ro, Jeju-si, Jeju Special Self-Governing Province 48015, Korea

[3] Department of Foodservice management, Youngsan University, 142 Bansongsunhwan-ro, Haeundae-gu, Busan 48015, Korea

\* Correspondence: syoon3@chu.ac.kr (S.Y.); shan@ysu.ac.kr (S.-H.H.); Tel.: +82-64-741-6514 (S.Y.); +82-51-540-7182 (S.-H.H.)

**Abstract:** The purpose of this study was to investigate the awareness of the necessity and importance of liberal arts education and to examine the satisfaction of college students with their liberal arts courses. This study was conducted from June 1–15, 2018, for college students who are taking liberal arts courses. The collected data were analyzed using the SPSS 24.0 and AMOS 24.0 statistical package programs. To understand the general characteristics of the survey subjects, a frequency analysis, exploratory factor analysis, correlation analysis, and reliability analysis were performed to measure the reliability and validity of the measurement tools, and a structural model analysis was conducted to verify the proposed research model. The result shows that a professor's professionalism has a positive influence on the perception of a subject's importance and necessity after the course, diversity has a positive influence on satisfaction in liberal arts education. Favorable changes in the perception of importance and necessity have a positive effect on satisfaction level. Our findings imply that colleges should operate an integrated student-selective education course that allows all students to select and take liberal arts courses. It should be organized to secure full-time professors who will be exclusively responsible for liberal arts curriculums.

**Keywords:** liberal arts education; professionalism; perceptions; diversity

## 1. Introduction

Amid the digital information revolution of the 21st century, general and character education, which have previously been neglected by 20th century college education's focus on cultivating professional intellectuals, have become the center of changing demand. It could be considered a natural response to the rapid economic development, and the change has had a variety of social, cultural, and political side-effects, emphasized by the competitive growth and social change in Korean society. Demand for new talent to meet the rapidly changing digital information age has also increased, which has of course led to a change for talent in college education in South Korea. The demand has grown for the positive talents of liberal education specializing in intellectual training for future needs such as collaboration with others, empathy, communication, consideration, and mutual respect.

According to the Ministry of Ministry of Science, ICT and Future Planning, the Ministry of Education and the Ministry of Labor announced a joint venture of South Korea government in August in 2018 in order to cultivate customized fusion talent for the creative industry. It proposed to conduct a

liberal arts-based convergence education so that the humanities' imagination can be projected into science and technology. This university is also the reason that the importance of basic general education is emphasized as well as character education and citizenship education through liberal art education in the humanities.

Liberal education is generally named in various terms such as liberal arts, basic education, liberal arts education, universal education, liberal arts education, general education, etc. [1].

It has been recognized as a preliminary study of humanities with weak academic character and lower levels compared to major education or basic education subjects. Liberal education is a concept that contrasts with major education and is similar to liberal arts education. Liberal education has its roots in civil liberties education in ancient Greece, which means the liberation from ignorance. It can also be understood as meaning free education that can be participated in by all citizens, and a civic education that teaches ethics, morality, values, norms, and art, that should mastered by citizens. Liberal education in modern college education began in 1945 with the report, "A General Education in Free Society", published by Harvard University in the United States. Today, liberal arts education in the United States has undergone three reforms of Harvard's liberal arts education. Students in the bachelor's course must have completed about 25% of their total credits in liberal arts courses. Its content consists of five areas: literature, art, history, society, science, and foreign culture [2].

Despite both professors and students recognizing the importance of liberal arts education, the satisfaction level of liberal arts education is low, and the demand for the improvement of liberal art education courses by means such as diversity in subjects is high [1,3]. According to a comparative study on the cognition of major subjects and liberal arts subjects, in the case of major subjects, the professionalism and enthusiasm of the professors, and in the case of liberal arts subjects, the self-directed competence and internal motivation of the students, were the main factors affecting the difference [4]. We can therefore assume that it is possible to raise awareness and satisfaction in class by targeting the delivery of liberal knowledge and emotional factors such as interest and the exploration of various liberal arts areas, students' interests, and emotions.

Rosenberg [5] used Harvard Magazine in his comments on research reports on instructors and students regarding Harvard University's liberal arts curriculum. It was pointed out that there is a tendency to select courses that are relatively easier to obtain good grade points in or classes that have less learning volume rather than considering interests and careers. Therefore, in a study for improving the quality of liberal arts education, it is necessary to conduct a survey of cognition and satisfaction from various perspectives on the needs of the consumers and learners of liberal arts education to give feedback to the curriculum.

Thompson, Eodice, and Tran [6] pointed out the reasons for students' low awareness of liberal education. The purpose of the courses were poorly communicated and it was unclear how it would help them make decisions about their career. In this regard, the professionalism of professors, such as their teaching methods and the quality of lectures, influences the improvement of awareness of the goals of a liberal arts curriculum for college students [7].

For talent in the era of the fourth industrial revolution, companies emphasize convergence and the ability to combine. However, the university pointed out that organizing its major-oriented curriculum is a problem [8]. The students should not only learn knowledge and skills in various fields such as humanities, social sciences, natural sciences, arts, and so on, but also arrange the courses so that they can be interconnected and understand them from an integrated perspective. It is necessary to design, develop, and utilize a diverse and systematic curriculum [9]. Thus, in the role of university education, education in liberal arts is urgently applied to understand and acquire the viewpoints and values of convergence and the sophisticated knowledge required by modern society.

For effective liberal arts education, class satisfaction, and academic achievement, it is meaningful to discuss the types of learning and gender differences among college students. This is because it is necessary to develop a teaching and learning method that takes into account the students' learning style and gender differences in the field of liberal arts education in South Korea. Previous studies have

found significant differences depend on genders about the importance and necessity of perception in education [1,4,7–11]. One of the studies showed that significant differences in learning styles by genders were statistically identified: female students preferred sensing, verbal, reflective, and sequential styles more than male students, while male students preferred intuitive, visual, active, and global learning styles more than female students [10]. Differences in satisfaction concerning the participation in liberal arts education classes were found by gender; male students showed higher average values than female students except for educational factors, which are educational and resting factors that show statistically significant differences [11].

The purpose of this study is to analyze the effects of diversity in the liberal arts curriculum and the professionalism of professors on the perception and satisfaction of liberal arts education. As liberal arts education is becoming strongly emphasized in universities, universities are reforming their various liberal arts education to improve the quality. Therefore, this study aims to investigate the effects of the questionnaires on the cognition and satisfaction of liberal arts education for students who take liberal arts education.

## 2. Hypothesis

**Hypothesis 1:** *A professor's professionalism has a positive influence upon the perception of liberal arts education.*

**Hypothesis 2:** *The diversity of the liberal arts education has a positive influence upon the perception of liberal arts education.*

**Hypothesis 3:** *A professor's professionalism has a positive influence upon the satisfaction of liberal arts education.*

**Hypothesis 4:** *The diversity of the liberal arts education has a positive influence upon the satisfaction of liberal arts education.*

**Hypothesis 5:** *The perception of liberal arts education has a positive influence upon the satisfaction of liberal arts education.*

## 3. Materials and Methods

### 3.1. Data Collection and Sampling

In the era of the 4th industrial revolution in the 21st century, the importance of liberal arts education and personality education has emerged. Therefore, not only four-year universities in Korea but also junior colleges are considering the common liberal arts education and expanding the range of operations. Therefore, this study examined students' perception and satisfaction related to liberal arts education.

This study was conducted for 15 days from June 1–15, 2018, for college students who are taking liberal arts courses at a university in Jeju, South Korea, where a liberal arts curriculum is operated. Five hundred copies of the questionnaire were distributed, 490 of which were collected, and 477 were used for analysis, leaving out 23 which were unsatisfactory. The collected data were analyzed using the SPSS 24.0 and AMOS 24.0 statistical package programs. In this study, SPSS statistical software was used to verify and correct variable items using the general characteristics of the survey subjects, a frequency analysis, an exploratory factor analysis, a correlation analysis, and a reliability analysis. This is because the SPSS statistical software can easily perform both a parametric and non-parametric comparative analysis compared to other programs [12]. In addition, a structural model analysis was performed using the AMOS program to examine the causal relationship between a number of independent and dependent variables [13,14]. This is because AMOS software uses the Maximum likelihood (ML) estimation technique in structural model analyses and is widely used to confirm theories [13,14].

Of the respondents, females represented 63.9%, and male represented 36.1%. First year students made up 92.7% of the total respondents, followed by 3.4% for second year, 3.1% for third year, and

0.8% for fourth year. By Major, nursing and health was the highest at 57.7%, tourism at 21.8%, social science at 11.3%, natural science at 5.0%, engineering at 2.5%, and art and physical education at 1.7%. Table 1 shows a summary of the respondents' demographic characteristics.

**Table 1.** Demographic characteristics of respondents (n = 477).

|  | **Category** | **Frequency** | **Percentage (%)** |
|---|---|---|---|
| Gender | Male | 172 | 36.1 |
|  | Female | 305 | 63.9 |
| Year of Students | First year | 442 | 92.7 |
|  | Second year | 16 | 3.4 |
|  | Third year | 15 | 3.1 |
|  | Fourth year | 4 | 0.8 |
| Majors | Nursing and health science | 275 | 57.7 |
|  | Hospitality and tourism | 104 | 21.8 |
|  | Social science | 54 | 11.3 |
|  | Arts and science | 24 | 5.0 |
|  | Engineering | 12 | 2.5 |
|  | Arts and physical education | 8 | 1.7 |

*3.2. Measures*

In order to measure each construct, multiple items were generated based on previous studies [15]. Measurements of the selected liberal arts curriculum were explained to liberal arts officers and related professors, and their opinions were combined to make the final selection. All items were measured using a five-point Likert scale ranging from "strongly disagree" to "strongly agree". Changes in perception and satisfaction were measured in a single question.

## 4. Results

*4.1. Analysis of Validity and Reliability*

In the exploratory factor analysis, a principal component analysis was used for the extraction of factors, and varimax rotation was applied for the rotation method. The criteria for the selection of factors and items were considered statistically significant when the eigenvalue was above 1.0 and the factor loading was above 0.5 [16]. As a result, two factors were derived, as shown in Table 2. The eigenvalue of the first factor was 3.664 (45.773%). The first factor included four variables: faithfulness of contents, passion for teaching, related experiences, and justification for evaluation. The first factor is called the professor's professionalism because it represents the teaching experience and teaching philosophy. The eigenvalue of the second factor was 1.886 (23.572%). The second factor included four variables: diversity of subjects, variety in teaching methods, quality of the education on the subject, and conformity of teaching materials. The second factor is called diversity because it explains the type of liberal arts education and the diversity of operations. Next, as a result of examining the exploratory factor analysis, the KMO value was 0.883 and Bartlett's Test of Sphericity was 2055.853 (df = 28) ($p < 0.001$), indicating that the samples used for the factor analysis were suitable.

All multiple items were subjected to reliability analyses. Reliabilities were assessed using Cronbach's $\alpha$ coefficient. As shown in Table 2, Cronbach's $\alpha$ coefficients for the constructs ranged from 0.755 to 0.891. Coefficients exceeding 0.70 were considered acceptable [17]. Table 3 presents correlations between the constructs, means, and standard deviations.

**Table 2.** Factor analysis and reliability analysis.

| Factor | Factor Loading | Eigen Value | Variance | Reliability |
|---|---|---|---|---|
| **Professor's professionalism** | | 3.662 | 45.773 | 0.891 |
| Faithfulness of contents | 0.894 | | | |
| Passion for teaching | 0.880 | | | |
| Related experiences | 0.821 | | | |
| Justification for evaluation | 0.755 | | | |
| **Diversity** | | 1.886 | 23.572 | 0.755 |
| Diverse content of subjects | 0.874 | | | |
| Variety of teaching method | 0.639 | | | |
| Quality of the educational subject | 0.581 | | | |
| Conformity of teaching materials | 0.514 | | | |

KMO = 0.883 Bartlett's Test of Sphericity = 2055.835 df = 28 p < 0.001.

**Table 3.** Results of correlations between constructs.

| | Mean | SD | Professionalism | Diversity | Perception | Satisfaction |
|---|---|---|---|---|---|---|
| Professionalism | 3.671 | 0.854 | 1 | | | |
| Diversity | 3.044 | 0.722 | 0.611** | 1 | | |
| Perception | 3.199 | 1.017 | 0.613** | 0.679** | 1 | |
| Satisfaction | 3.296 | 1.008 | 0.351** | 0.620** | 0.574** | 1 |

** $p < 0.01$ (two-tailed test).

### 4.2. Structural Model and Hypotheses Testing

To measure each construct, multiple items were generated. Maximum-likelihood estimates of the various parameters of the model are given in Table 4 and Figure 1. Th overall evaluation of the model fit was based on multiple indicators [17]. Although the chi-square statistic suggests that the data does not fit the model well ($\chi^2$ = 127.959, df = 33; p< 0.001), all other overall model fit indicators suggest that the data fit the model well: GFI = 0.950, AGFI = 0.971, NFI = 0.953, CFI = 0.965, and RMR = 0.050. The squared multiple correlation (SMC; $R_2$) statistics of the structural equations for perception and satisfaction are 0.540 (54.0%) and 0.510 (51.0%), respectively.

**Table 4.** Standardized structural estimates.

| | Path | Standardized Coefficients | C.R. | P | Result |
|---|---|---|---|---|---|
| $H_1$ | Professionalism → Perception | 0.110 | 1.764 | 0.078* | Supported |
| $H_2$ | Diversity → Perception | 0.649 | 9.017 | 0.000** | Supported |
| $H_3$ | Professionalism → Satisfaction | −0.376 | −5.575 | 0.000** | Not supported |
| $H_4$ | Diversity → Satisfaction | 0.771 | 8.443 | 0.000** | Supported |
| $H_5$ | Perception → Satisfaction | 0.228 | 4.022 | 0.000** | Supported |
| | SMC ($R^2$) | | | | |
| | Cognition | 0.542(54.2%) | | | |
| | Satisfaction | 0.510(51.0%) | | | |
| | $\chi^2$ | 48.793 | | | |
| | df | 23 | | | |
| | p | 0.001 | | | |

$\chi^2$ = 127.959, df = 33, p = 0.000 ($\chi^2$/df = 3.878), GFI = 0.950, AGFI = 0.971, NFI = 0.953, CFI = 0.965, RMR = 0.050; *** $p < 0.01$, ** $p < 0.05$, * $p < 0.1$.

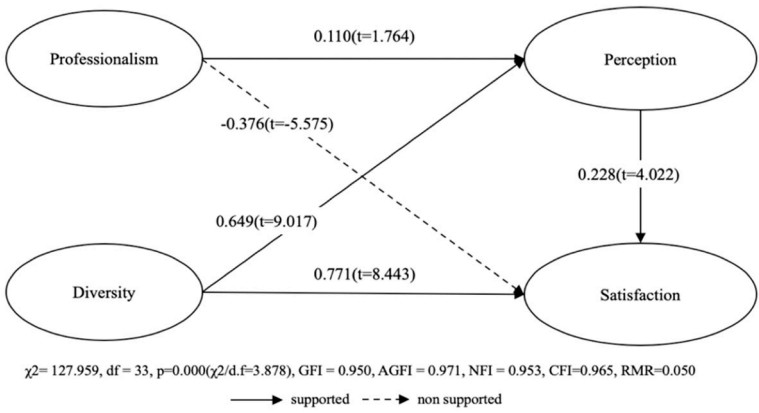

χ2= 127.959, df = 33, p=0.000(χ2/d.f=3.878), GFI = 0.950, AGFI = 0.971, NFI = 0.953, CFI=0.965, RMR=0.050

——▶ supported   - - -▶ non supported

**Figure 1.** Research model result.

The result shows that professionalism has a positive effect on perception ($\gamma$ = 0.110, p > 0.05), supporting $H_1$, and diversity has a positive impact on perception ($\gamma$ = 0.649, p < 0.001), supporting $H_2$. The $H_3$ hypothesis test showed that professionalism has no positive effect on satisfaction ($\gamma$ = −0.376, p < 0.001). Therefore, hypothesis $H_3$ was not supported. The study results support $H_4$ and $H_5$, because diversity ($\gamma$ = 0.771, p < 0.001) and perception ($\gamma$ = 0.228, p < 0.001) have a positive effect on satisfaction.

### 4.3. Measurement Equality Analysis

The measurement equality was examined to determine whether there was a significant difference in the effects of the professor's professionalism and the diversity on satisfaction and perception. Depending on the research field, if at least one level or if level 1 and level 2 are met, then the measurement identity may be considered to be satisfied [18]. Measurement identity verification is achieved by verifying the differences between two groups of constrained and non-constrained models [19,20]. As shown in Table 5, the measurement equality analysis, factor loading constraint model (model 3, $\Delta\chi^2$ = 6.249, p > 0.05), covariance constraint model (model 4, $\Delta\chi^2$ = 7.622, p > 0.05), factor loading, and covariance and error variance constraint model (model 5, $\Delta\chi^2$ = 5.704, p > 0.05) are significantly statistically different from the non-constrained; however, the factor loading and covariance constraint model (model 4) showed no statistically significant difference to the non-constrained model (model 4, $\Delta\chi^2$ = 43.920, p < 0.01). Although there was a statistical difference in Model 4, it was determined that there was no problem in measurement equality because there was no statistical difference in the other non-constrained models.

**Table 5.** A result of measurement equality analysis.

|  | $\chi^2$ | df | GFI | CFI | RMSEA | TLI | $\Delta\chi^2$ | Sig. |
|---|---|---|---|---|---|---|---|---|
| Nonconstrained Model | 99.804 | 38 | 0.952 | 0.970 | 0.059 | 0.956 | - | - |
| λ Constrained Model | 106.053 | 44 | 0.949 | 0.970 | 0.054 | 0.962 | 6.249 | not supported |
| φ Constrained Model | 107.426 | 47 | 0.948 | 0.971 | 0.052 | 0.965 | 7.622 | not supported |
| λ, φ Constrained Model | 143.724 | 55 | 0.932 | 0.957 | 0.058 | 0.957 | 43.920 | Supported |
| λ, φ, θ Constrained Model | 105.507 | 41 | 0.950 | 0.969 | 0.058 | 0.958 | 5.704 | not supported |

### 4.4. Gender Difference Analysis

The analysis was divided into male and female groups to examine the influence of the professionalism and diversity of liberal arts curriculum on both groups regarding perception and satisfaction; the analysis was divided into male and female groups. The results are shown in Table 6, Figure 2.

**Table 6.** Standardized structural estimates by group.

|  | Path | Male (C.R.) | Female (C.R.) |
|---|---|---|---|
| $H_1$ | Professionalism → Perception | 0.267(2.375) *** | 0.051(0.693) |
| $H_2$ | Diversity → Perception | 0.734(3.989) *** | 0.712(7.922) *** |
| $H_3$ | Professionalism → Satisfaction | −0.293(−2.492) ** | −0.393(−4.733) *** |
| $H_4$ | Diversity → Satisfaction | 0.813(5.738) *** | 0.701(6.053) *** |
| $H_5$ | Perception → Satisfaction | 0.157(1.801) * | 0.283(3.813) *** |

***$p < 0.01$, ** $p < 0.05$, * $p < 0.1$.

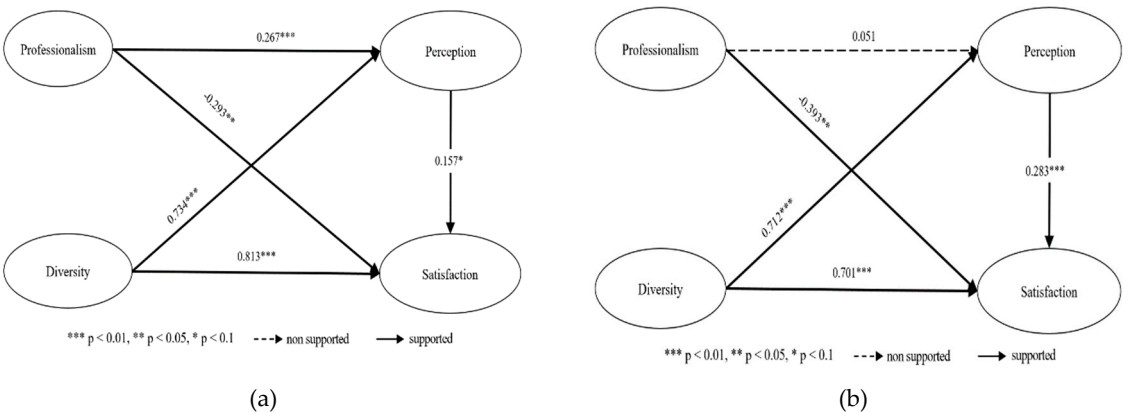

(a)              (b)

**Figure 2.** The path analyses by group. (**a**) Male group; (**b**) female group.

In general, professionalism had a negative effect on satisfaction, which might be interpreted as, in liberal arts, professors should deliver general and practical internal information rather than professional knowledge. Also, the professor's professionalism was found to have a positive influence on the perceptions of men, but not of women. However, these results only show the significance of each group's paths and do not explain the differences in paths between the two groups. Therefore, a multi-group analysis was used to verify significant path differences between the two groups. The results are shown in Table 7. According to the analysis results of Tables 6 and 7, the following results were obtained. In the case of male students, the diversity of liberal arts had more influence on their perception than it did on the female students.

**Table 7.** Results of multi-group path constraints.

| Path Constraints | $\chi^2$ | df | $\Delta\chi^2$ | Sig. |
|---|---|---|---|---|
| Professionalism → Perception | 173.183 | 67 | 1.713 | N/A |
| Diversity → Perception | 175.692 | 67 | 4.222** | Supported |
| Professionalism → Satisfaction | 173.276 | 67 | 1.806 | N/A |
| Diversity → Satisfaction | 171.723 | 67 | 0.253 | N/A |
| Perception → Satisfaction | 172.214 | 67 | 0.744 | N/A |

Constraint model: $\chi^2$ = 171.470, df = 66, p = 0.000($\chi^2$/df = 2.598), GFI = 0.934, AGFI = 0.891, NFI = 0.939, CFI = 0.961, RMR = 0.050. ***$p < 0.01$, ** $p < 0.05$, * $p < 0.1$.

## 5. Discussion

This study analyzed whether there was a change in the perception of the need and importance of liberal arts education after taking a liberal arts curriculum, for 477 students at a university in Jeju, South Korea. This study examined the impact of the diversity of the liberal arts curriculum on the satisfaction of the liberal arts curriculum currently run by the university. The results of this study can be summarized as follows.

The professor's professionalism has a positive effect (+) on the change in the perception of the importance and necessity of liberal arts education, and diverse subjects have a positive effect (+) on the change in the perception and educational satisfaction. Furthermore, the change in the perception of the importance and necessity of liberal arts education after completing the curriculum has a positive effect (+) on the satisfaction of liberal arts education. Compared to previous studies, most indicators show similar results [1,4,7–9,21,22].

In this study, in particular, the change in importance and awareness of liberal arts education is positively impacted by a professor who shows professionalism and passion, faithfulness to the contents of the class, a passion for teaching, related experiences, and justification for evaluation [4,21]. In addition, the results of our study found that the professor's professionalism did not affect satisfaction. On the other hand, another result of our research shows that the importance and need for satisfaction through the change in perception with the liberal arts education had a positive effect if the professionalism of the professor and their passion for liberal arts education were high. In addition, the higher the diversity of subjects, variety of teaching methods, quality of the educational subjects, and degree of conformity of teaching materials, the higher the satisfaction of the liberal arts curriculum. However, in this study, it was found that the professor's professionalism and passion for liberal arts did not appear to affect the liberal arts education satisfaction.

The findings are similar to the results of another study, which showed that students are somewhat aware of the importance of liberal arts education [1,9]. This can be interpreted as a recognition that students' awareness of the importance of liberal arts education, which is the result of this study, was high after the course, but not as important as the major subjects [4].

According to [7,21], students often perceive liberal arts courses as a compulsory lower-level study for their degree. The fact that there is a lack of awareness or support for this study can explain why the survey subjects were primarily first year and second year students. Additionally, according to the research results of Son [22], most of the students recognized the need for liberal arts education in general. Also, the demand for the establishment of various courses was high. This is similar to the fact that various combinations of liberal arts in this study affect educational satisfaction.

It is probably natural that there were no significant differences between male and female students in this study. The liberal arts curriculum in South Korea consists of 40–60 students who are lectured in large units, and classes are predominantly teacher-centered. If you continue to apply the instructor-oriented lecture-based method for delivering knowledge to male and female students who prefer a variety of learning styles in liberal arts, the satisfaction and learning effect of the class will appear low regardless of gender. Professors should recognize the students' individual learning styles and gender differences, then reflect these in their classes to enhance students' learning effectiveness. To provide a learner-centered learning environment and to develop effective teaching strategies and teaching and learning methods, institutional and financial support from the university authorities, continuous research by professors, and active academic follow-up will be required.

The generalization of the results of this study to all universities in South Korea has limitations, due to the study being conducted in a single university. Long-term research is required to make the liberal arts curriculum of specific universities work, change students' perceptions, and improve their satisfaction with their courses.

The study of the students' perception survey on the existing liberal arts education was mainly conducted by the variance analysis [15]. However, this study is significant in that it investigated the relationship between the recognition of liberal arts education, change in perception, and satisfaction of students. In addition, it is significant that the students' perception of liberal arts education was divided into the specialized knowledge of the professor and the diversity of the subjects. Finally, there are theoretical and practical implications for verifying the effectiveness of the liberal arts education through an empirical analysis of the effect of the change in students' perception and satisfaction.

The importance and necessity of liberal arts education are emphasized and used as basic data to transform liberal arts education in universities. We suggest that, based on the operational model of the

liberal arts education curriculum that reflects ongoing evaluation and feedback, we will be able to carry out personalized education for the liberal arts with continuous participation and development.

**Author Contributions:** Conceptualization, M.-Y.A., S.Y. and S.-H.H.; data curation, S.Y. and S.-H.H.; formal analysis, M.-Y.A., S.Y. and S.-H.H.; methodology, M.-Y.A., S.Y. and S.-H.H.; project administration, M.-Y.A., S.Y.; supervision, M.-Y.A.; visualization, S.Y. and S.-H.H.; writing—original draft, M.-Y.A., S.Y. and S.-H.H.; writing—review & editing, M.-Y.A., S.Y. and S.-H.H. All authors have read and agreed to the published version of the manuscript.

**Funding:** This research received no external funding.

**Conflicts of Interest:** The authors declare no conflict of interest.

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
