# Peer review of "The Effects of a Professor’s Professionalism and Diversity on the Perception and Satisfaction of Education in the Liberal Arts Curriculum"

_sustainability, doi:10.3390/su12093689_

Round 1
Reviewer 1 Report
The study was conducted correctly and limitations were noted. The results are significant for further research on the stauts and value of arts in education.
Author Response
Dear. reviewer
Thank you for giving me the opportunity to submit a revised draft of my manuscript tilted The Effects of Professor's Professionalism and Diversity on the Perception and Satisfaction of Education in the Liberal Arts Curriculum to Sustainability. We appreciate the time and effort that you have dedicated to providing your valuable feedback on our manuscript. We are grateful to the reviewers for their insightful comments on our paper. We have been able to incorporate changes to reflect most of the suggestions provided by the reviewers. We have highlighted the changes within the manuscript.
In addition to the above comments, all spelling and grammatical errors pointed out by the reviewers have been corrected.
We look forward to hearing from you in due time regarding our submission and to respond to any further questions and comments you may have.
Sincerely,
Mi Young An, Susie Yoon, and Sang Ho Han
Reviewer 2 Report
The article shows a study of great interest and topicality covering the whole of a research paper section. The methodology is well thought out as well as the development of the analysis and the presentation of results. The conclusions reflect the contrast of the hypotheses and the formal aspects are strictly followed.
Author Response
Dear. reviewer Thank you for giving me the opportunity to submit a revised draft of my manuscript tilted The Effects of Professor's Professionalism and Diversity on the Perception and Satisfaction of Education in the Liberal Arts Curriculum to Sustainability. We appreciate the time and effort that you have dedicated to providing your valuable feedback on our manuscript. We are grateful to the reviewers for their insightful comments on our paper. We have been able to incorporate changes to reflect most of the suggestions provided by the reviewers. We have highlighted the changes within the manuscript. Here is a point-by-point response to your comments and concerns. |
|
Comment 1 |
The article shows a study of great interest and topicality covering the whole of a research paper section. The methodology is well thought out as well as the development of the analysis and the presentation of results. The conclusions reflect the contrast of the hypotheses and the formal aspects are strictly followed. |
Response 1 |
Thank you for pointing this out. We agree with this comment. Therefore, we have changed the interpretation that the professor's professionalism influenced the change in perception from a significance level of 0.1 to a statistically significant, in order to the professors’ professionalism needs to be more emphasized. Also, we’ve got rid of the contrast hypotheses in conclusion. (Line 197-201)
In addition to the above comments, all spelling and grammatical errors pointed out by the reviewers have been corrected. We look forward to hearing from you in due time regarding our submission and to respond to any further questions and comments you may have. Sincerely, Mi Young An, Susie Yoon, and Sang Ho Han |
Reviewer 3 Report
This subject was of great interest to me and I think it is a worth while study. However, there are some problems with the use of English which means that communication is not effective.
On lines 32 and 33, 'universal intelligence education' does not make sense do you mean 'basic general education'? Rather than 'personality education' do you mean 'character education'? there are other examples where the meaning is not clear.
I recommend looking at the introduction again to improve the English. I would also be clear about the study being conducted in one University from the start of the paper, this was only made clear in the discussion part of the paper.
The liberal arts curriculum (pages 45-55) can be described as a fundamental or foundation course comprising of a range of subjects, however, I would say that it is not necessarily 'basic' which implies a low level, I suppose this depends on the wider educational context in which the curriculum is being taught.
Although the method is quite well explained, I would like to know more about the questionnaire design. How did the researchers decide which questions to ask? Why was the professor's professionalism and diversity identified as themes?
When looking at the results it seems that Diversity has a big impact on perception and satisfaction yet this is not mentioned in the discussion section.
Lines 233-236 seems to contradict each other. First it is stated that professor professionalism had a positive impact on satisfaction, then the paragraph ends with a sentence saying it had no effect on satisfaction.
I think some of these problems could sorted out when the English is improved.
Author Response
Dear. reviewer Thank you for giving me the opportunity to submit a revised draft of my manuscript tilted The Effects of Professor's Professionalism and Diversity on the Perception and Satisfaction of Education in the Liberal Arts Curriculum to Sustainability. We appreciate the time and effort that you have dedicated to providing your valuable feedback on our manuscript. We are grateful to the reviewers for their insightful comments on our paper. We have been able to incorporate changes to reflect most of the suggestions provided by the reviewers. We have highlighted the changes within the manuscript. Here is a point-by-point response to your comments and concerns. |
|
Comment 1 |
This subject was of great interest to me and I think it is a worth while study. However, there are some problems with the use of English which means that communication is not effective. On lines 32 and 33, 'universal intelligence education' does not make sense do you mean 'basic general education'? Rather than 'personality education' do you mean 'character education'? there are other examples where the meaning is not clear. |
Response 1 |
We agree with this comment. Therefore, according to the above comments, all spelling and grammatical errors revised.
|
Comment 2 |
I recommend looking at the introduction again to improve the English. I would also be clear about the study being conducted in one University from the start of the paper, this was only made clear in the discussion part of the paper. |
Response 2 |
Thank you for pointing this out. We have modified to clarify this point. Since this study was conducted in a single, university, we changed to clarify this point. This study was conducted for 15 days from June 1-15, 2018, for college students who are taking liberal arts courses at the A university in Jeju, South Korea, where liberal arts curriculum is operated. (Line 133-134) |
Comment 3 |
The liberal arts curriculum (pages 45-55) can be described as a fundamental or foundation course comprising of a range of subjects, however, I would say that it is not necessarily 'basic' which implies a low level, I suppose this depends on the wider educational context in which the curriculum is being taught. |
Response 3 |
Yes, it is. 'Basic' was intentionally added to emphasize that liberal arts education is basic education, but it was discussed that there is a possibility of misunderstanding after discussion. |
Comment 4 |
Although the method is quite well explained, I would like to know more about the questionnaire design. How did the researchers decide which questions to ask? Why was the professor's professionalism and diversity identified as themes? |
Response 4 |
We agree with this and have incorporated your suggestion throughout the manuscript.
The composition and selection process of the questionnaire was explained.
We explained the meaning of the factors by talking about the meaning of the questions for each factor.
|
Comment 5 |
When looking at the results it seems that Diversity has a big impact on perception and satisfaction yet this is not mentioned in the discussion section. Lines 233-236 seems to contradict each other. First it is stated that professor professionalism had a positive impact on satisfaction, then the paragraph ends with a sentence saying it had no effect on satisfaction. |
Response 5 |
Thank you for pointing this out. We agree with this comment. Therefore, we have changed the interpretation that the professor's professionalism influenced the change in perception from a significance level of 0.1 to a statistically significant, in order to the professors’ professionalism needs to be more emphasized. Also, we’ve got rid of the contrast hypotheses in conclusion. (Line 197) |
Comment 6 |
I think some of these problems could sorted out when the English is improved. |
Response 6 |
In addition to the above comments, all spelling and grammatical errors pointed out by the reviewers have been corrected. We look forward to hearing from you in due time regarding our submission and to respond to any further questions and comments you may have. Sincerely, Mi Young An, Susie Yoon, and Sang Ho Han |
Reviewer 4 Report
Dear Authors,
Thank you for your interesting study.
Below you will find some few suggestions for improving the manuscript.
INTRODUCTION: A competent introduction should include at least: 1) significance of the topic, 2) the information gap in the available literature associated with the topic, 3) a literature review, 4) subsequently developed purposes/objectives and hypotheses. So, it is important that you include a section with a literature review.
MATERIALS AND METHODS: The purpose of sufficient detail in the methods section is so that an appropriately trained person would be able to replicate your experiments. Please, review this section and provide more details about the methodology. Why liberal arts has been chosen? What is the allocation of the college students? Advantages and problems when using SPSS and AMOS?
DISCUSSION: Please discuss where your RESULTS are similar or different from other published evidence and why this might be so. What was different in methods or analysis, what was similar?
Author Response
Dear. reviewer Thank you for giving me the opportunity to submit a revised draft of my manuscript tilted The Effects of Professor's Professionalism and Diversity on the Perception and Satisfaction of Education in the Liberal Arts Curriculum to Sustainability. We appreciate the time and effort that you have dedicated to providing your valuable feedback on our manuscript. We are grateful to the reviewers for their insightful comments on our paper. We have been able to incorporate changes to reflect most of the suggestions provided by the reviewers. We have highlighted the changes within the manuscript.
In addition to the above comments, all spelling and grammatical errors pointed out by the reviewers have been corrected. We look forward to hearing from you in due time regarding our submission and to respond to any further questions and comments you may have. Sincerely, Mee Young An, Susie Yoon, and Sang Ho Han |
|
INTRODUCTION: |
A competent introduction should include at least: 1) significance of the topic |
Response 1 |
Thank you for pointing this out. We agree with this and have incorporated your suggestion throughout the manuscript. “Amid the digital information revolution of the 21st century, general and character education, which have previously been neglected by 20th century college education’s focus on cultivating professional intellectuals, became the center of changing demand. It could be considered a natural response to rapid economic development, and changing a variety of social, cultural and political side-effects emphasized by competitive growth and social change in Korea Society. Demand for new talent to meet the rapidly changing digital information age also increased, which of course led to a change for talent with a college education in Korea. The demand grew for positive talents of liberal education specializing in intellectual training for future needs such as collaboration with others, empathy, communication, consideration, mutual. ” (Line 36-44) |
Comment 2 |
2) the information gap in the available literature associated with the topic |
Response 2 |
Thank you for pointing this out. We agree with this and have incorporated your suggestion throughout the manuscript. “According to the Ministry of Science, ICT and Future Planning of the Government of South Korea in August 2013, the Ministry of Education of and the Ministry of Labor in South Korea announced a joint in order to cultivate customized fusion talent for the creative industry. It proposed to conduct a liberal arts-based convergence education so that the humanities' imagination can project into science and technology. This university is also the reason that the importance of basic general education is emphasized as well as character education and citizenship education through liberal art education in the humanities.”(Line 45-51) |
Comment 3 |
3) a literature review |
Response 3 |
Thank you for pointing this out. We agree with this and have incorporated your suggestion throughout the manuscript.
Despite both professors and students recognizing the importance of liberal arts education, the satisfaction level of liberal arts education is low, and the demand for improvement of liberal art education courses such as diversity in subjects is high [1,3]. According to a comparative study on cognition of major subjects and liberal arts subjects, in the case of major subjects, the professionalism and enthusiasm of the professors, and in the case of liberal arts subjects, self-directed competence and internal motivation of the students were the main factors affecting the difference [4]. We can therefore assume that it is possible to raise awareness and satisfaction in class when targeting the delivery of liberal knowledge and emotional factors such as interest and exploration of various liberal arts areas, students' interests, and emotions. Rosenberg [5] used Harvard Magazine in his comments on research reports on instructors and students regarding Harvard University's liberal arts curriculum. It was pointed out that there is a tendency to select courses that are relatively easier to obtain good grade points or classes that have less learning volume than considering interests and careers. Therefore, in the study for improving the quality of liberal arts education, it is necessary to conduct a survey of cognition and satisfaction from various perspectives on the needs of the consumer and learners of liberal arts education to reflect the feedback to the curriculum. Thompson, Eodice, and Tran [6] pointed out that the reasons for students' low awareness of liberal education. The purpose of the courses were poorly communicated and it was unclear how it would help them make decisions about their career. In this regard, the professionalism of professors such as their teaching methods and the quality of lectures influences the improvement of awareness for the goals of liberal arts curriculum for college students [7]. As talent in the era of the fourth industrial revolution, companies emphasize convergence and the ability to combine. However, the University pointed out that organizing its major-oriented curriculum is a problem [8]. The students should not only learn knowledge and skills in various fields such as humanities, social sciences, natural sciences, arts, and so on, but also arrange the courses so that they can be interconnected and understand them from an integrated perspective. It is required to design, develop, and utilize diverse and systematic curriculum [9]. Thus, in the role of university education, education of liberal arts is urgently needed to understand and acquire the viewpoints and values of convergence and sophisticated knowledge required by modern society. For effective liberal arts education, class satisfaction, and academic achievement, it is meaningful to discuss the types of learning and gender differences among college students. This is because it is necessary to develop a teaching and learning method that takes into account the students' learning style and gender differences in the field of liberal arts education in South Korea. Previous studies have found significant differences in gender students about the importance and necessity of cognition in education, while others have not [1][4][7][8][9][10]. One of the studies showed that significant differences in learning styles by genders were statistically identified: female students prefer sensing, verbal, reflective, and sequential styles more than male students, while male students prefer intuitive, visual, active, and global learning styles more than female students [10]. As a result of finding satisfaction in the participation in liberal arts education classes by gender, male students showed higher average values than female students except for educational factors, which are educational and resting factors that show statistically significant differences [11].” (Line 52-116) |
Comment 4 |
4) subsequently developed purposes/objectives and hypotheses |
Response 4 |
We agree with this and have incorporated your suggestion throughout the manuscript.
|
MATERIALS AND METHODS: Comment 1 |
Why liberal arts has been chosen? |
Response 1 |
We agree with this and have incorporated your suggestion throughout the manuscript. “In the era of the 4th industrial revolution in the 21st century, the importance of liberal arts education and personality education emerged. Therefore, not only four-year universities in Korea, but also junior colleges are considering the common liberal arts education and expanding the range of operations. Therefore, this study examined students' perception and satisfaction related to the liberal arts education.” (Line 128-132) |
Comment 2 |
What is the allocation of the college students? |
Response 2 |
Thank you for pointing this out. We have already described the target students in the text. Please note. This study was conducted for 15 days from June 1-15, 2018, for college students who are taking liberal arts courses at the A university in Jeju, South Korea, where liberal arts curriculum is operated.” (Line 133-134) |
Comment 3 |
Advantages and problems when using SPSS and AMOS? |
Response 3 |
Thank you for pointing this out. We applied your suggestion. “This study was conducted for 15 days from June 1-15, 2018, for college students who are taking liberal arts courses at the A university in Jeju, South Korea, where liberal arts curriculum is operated. 500 copies of the questionnaire were distributed, 490 of which were collected, and 477 were used for analysis, except for 23, which were unsatisfactory. The collected data were analyzed using SPSS 24.0 and AMOS 24.0 statistical package programs. In this study, SPSS statistical software was used to verify and correct variable items using general characteristics of the survey subjects, frequency analysis, exploratory factor analysis, correlation analysis, and reliability analysis. This is because the SPSS statistical software can easily perform both parametric and non-parametric comparative analysis compared to other programs [12]. In addition, structural model analysis was performed using the AMOS program to examine the causal relationship between a number of independent and dependent variables [13, 14]. This is because AMOS software uses the ML estimation technique in structural model analysis and is widely used to confirm the theory [13, 14].” (Line 133-144) |
DISCUSSION: Comment 1 |
Please discuss where your RESULTS are similar or different from other published evidence and why this might be so. |
Response 1 |
Thank you for pointing this out. We agree with this and have incorporated your suggestion throughout the manuscript.
In this study, in particular, the change of importance and awareness of liberal arts education are positively impacted by a professor who shows professionalism and passion, the faithfulness to the contents of the class, passion for the teaching, related experiences, and justification for evaluation[4][21]. In addition, the result of our study was found that the professor’s professionalism did not affect satisfaction. On the other hand, another result of our research shows that the importance and need for satisfaction through the change of perception with the liberal arts education had a positive effect if the professionalism of the professor and the passion for liberal arts education were high. In addition, the higher the diversity of subjects, variety of teaching methods, quality of the educational subjects, and degree of conformity of teaching materials, the higher the satisfaction of the liberal arts curriculum. However, in this study, it was found that the professor's professionalism and passion for liberal arts did not appear to affect liberal arts education satisfaction. The findings are similar to the results of another study, which showed that students are somewhat aware of the importance of liberal arts education [1][9]. This can be interpreted as a recognition that students' awareness of the importance of liberal arts education, which is the result of this study, was high after the course, but not as important as the major subjects [4]. According to [7][21], students are often perceived liberal arts courses as a compulsory lower-level study for the degree. The fact that there is a lack of awareness or support for this study can be explain why the survey subjects are primarily first year and second year students. Additionally, according to the research results of Son [22], most of the students recognized the need for liberal arts education in general. Also, the demand for the establishment of various courses was high. This is similar to the fact that various combinations of liberal arts in this study affect educational satisfaction. It is probably natural that there were no significant differences between male and female students in this study. The liberal arts curriculum in South Korea consist of 40-60 students who are lectured in large units, and classes are predominantly teacher-centered. If you continue to apply the instructor-oriented lecture-based method for delivering knowledge to male and female students who prefer a variety of learning styles in liberal arts, the satisfaction and learning effect of the class will appear low regardless of gender. Professors should recognize the students’ individual learning styles and gender differences, then reflect these in their classes to enhance students’ learning effectiveness. To provide a learner-centered learning environment and to develop effective teaching strategies and teaching and learning methods, institutional and financial support from the university authorities, continuous research by professors, and active academic follow-up will be required.”(Line 262-298) |
Comment 2 |
What was different in methods or analysis, what was similar? |
Response 2 |
Thank you for pointing this out. We agree with this and have incorporated your suggestion throughout the manuscript.
|
Round 2
Reviewer 3 Report
There has clearly been a lot of work done on this article. I was really pleased to see that it reads a lot better. The information is very clear. The contextual information at the beginning was well communicated. The authors have done a good job in responding to the feedback.
Reviewer 4 Report
Thanks to provided a new version and a covering letter.
I recommend accept the paper in present form.